# APR: Asymmetric Projector Regularization Improves EEG-to-Vision Contrastive Learning

## Abstract

Contrastive learning approaches that align electroencephalography (EEG) with visual representations have made progress in recent years, yet performance remains limited under the strict 200-way zero-shot setting. We propose APR (Asymmetric Projector Regularization), a strategy that imposes asymmetric capacity on the dual-tower alignment. Concretely, the EEG branch uses a single minimal linear head followed by $\ell_2$ normalization, while the image branch retains a small, learnable projector (linear or two-layer MLP) to accomplish the necessary geometric alignment. Built on NICE-style contrastive training with a frozen CLIP visual encoder, APR lifts zero-shot Top-1 accuracy from 13.8% to 30.45%, with concurrent gains in Top-5, mean rank (mRank), and mean reciprocal rank (MRR). Systematic ablations show that simplifying the EEG-side projector improves generalization, whereas over-simplifying the image-side projector causes substantial performance degradation. APR remains robust across subjects, time windows, and batch size or temperature settings. We advocate for this small EEG head plus small, learnable image head recipe as a simple, reusable practice for more reliable cross-modal alignment in high-noise, low-sample EEG-to-vision scenarios. Code available at https://anonymous.4open.science/r/only-for-test-2026.

## 1 Introduction

Decoding visual perception from noninvasive EEG is both compelling and difficult. EEG provides millisecond temporal resolution and wide accessibility, but suffers from a low signal-to-noise ratio, subject variability, and limited data. Early "brain-to-image" studies explored generative pipelines: either directly reconstructing images from EEG with GANs (Goodfellow et al., 2014) or diffusion, or distilling EEG into a teacher model for semantic alignment. These works validated the feasibility of EEG-driven visual reconstruction under few-shot conditions, but the fidelity of the generation and semantic consistency remained strongly constrained by the data scale (Singh et al., 2023).

Recent progress has leveraged large-scale generative priors. For instance, guided diffusion models use EEG embeddings as high-level semantic priors to produce higher-fidelity reconstructions, often incorporating adaptive EEG encoders such as the Adaptive Thinking Mapper (ATM) to condition multi-stage diffusion (Li et al., 2024). Beyond vision alone, multimodal approaches such as BraVL integrate language as a third modality. Using a brain-visual-linguistic mixture-of-products-of-experts (MoPoE) framework with mutual information regularization, BraVL learns joint latent variables that generalize in low-data or semi-supervised regimes, mitigating the semantic sparsity of EEG–vision alignment (Du et al., 2023). Similarly, NICE benchmarks confirm that adding language priors consistently improves semantic supervision (Song et al., 2024). These advances have been catalyzed by richer datasets such as the THINGS-EEG series (Ferrante et al., 2024; Bai et al., 2025), which standardize evaluation across subjects and categories.

Despite these advances, one design degree of freedom remains underexplored: how should projection heads be configured in EEG–vision dual-tower contrastive learning? Most works adopt symmetric or heuristic designs, overlooking EEG's noise sensitivity and the anisotropy of frozen vision spaces such as CLIP (Radford et al., 2021).

In this study we address this gap with Asymmetric Projection Regularization (APR). Through systematic ablation, we find that freezing CLIP while simplifying the EEG-side head to a minimal linear layer (Linear + $\ell_2$ normalization) yields large improvements in zero-shot recognition. Meanwhile, the vision side still requires a small but learnable head (linear or shallow MLP) to absorb alignment and scale calibration. In addition to NICE's contrastive training, APR increases the accuracy of the 200-way zero-shot Top-1 from 13. 8% to 30. 45%, with consistent gains in the Top-5 and ranking metrics (mRank/MRR). Our analysis further shows that:

- Over-parameterized EEG heads tend to memorize noise, degrading generalization.
- A minimal yet trainable vision head stabilizes alignment; over-simplification collapses inter-class margins.

Our main contributions can be summarized as follows:

1. New perspective on projection design: We are the first to systematically analyze projection head asymmetry in EEG–vision contrastive learning, and propose APR with reproducible implementation (EEG: 1-layer linear; Vision: shallow linear/MLP).

2. Strong performance gains: APR boosts 200-way zero-shot recognition from 13.8% to 30.45%, with substantial improvements across retrieval metrics.

3. Mechanistic insight: Through controlled ablations, we show why asymmetric projection matters: vision-side learnability supports geometric alignment, while EEG-side depth mainly amplifies noise.

In summary, APR introduces a simple yet effective modification that significantly improves EEG–vision zero-shot decoding. Our findings suggest a broader principle: when aligning noisy, low-resource modalities with frozen large-scale vision encoders, asymmetry in projection capacity can be key to both stability and generalization.

## 2 RELATED WORK

*Generative EEG-to-Image Reconstruction.* Early works approached EEG decoding with generative models. EEG2IMAGE introduced a two-stage pipeline -contrastive learning for EEG representation, followed by conditional GAN synthesis—to reconstruct 128×128 images from few-shot EEG. This validated the feasibility of brain-to-vision reconstruction, but image quality and semantic fidelity remained limited by the data scale (Singh et al., 2023). More recent methods leverage diffusion priors: Li et al. proposed an adaptive EEG encoder (ATM) aligned with CLIP, and employed a two-stage guided diffusion process to refine reconstructions, achieving strong zero-shot retrieval and classification (Li et al., 2024). Extensions based on knowledge distillation and latent diffusion (Ferrante et al., 2024) or multimodal priors such as DreamDiffusion (Bai et al., 2025; 2023) further demonstrated the robustness of diffusion-guided reconstruction.

*Contrastive Learning Baselines.* In parallel, contrastive learning has established reproducible benchmarks. NICE-EEG proposed a dual-tower architecture (EEG encoder + image encoder) with InfoNCE alignment, reporting 200-way zero-shot retrieval (Top-1 $\approx$ 15.6%) and systematically analyzing time window, channel, and semantic effects (Song et al., 2024). The subsequent work explored improved crossmodal consistency (Chen et al., 2024), asymmetric projection strategies (Mai et al., 2024) , and projection head calibration (Chen et al., 2025-06-25), collectively highlighting the role of projection capacity in generalization.

*Language-augmented Decoding.* Beyond vision-only pipelines, BraVL integrated language as a third modality via a brain-visual-linguistic MoPoE framework with mutual information regularization. This improved joint latent representations, enabling semi-supervised training with unpaired data and outperforming any single/double modality setting (Du et al., 2023). Similar trends appear in MindReader (Shukla, 2024) and Necomimi (Chen, 2024), highlighting language priors as semantic anchors for EEG–vision alignment.

*Open Gap.* Overall, the field has progressed from GAN $\rightarrow$ diffusion $\rightarrow$ contrastive learning $\rightarrow$ multimodal fusion. However, an overlooked degree of freedom is projection head asymmetry in dual-tower alignment. Most works assume symmetric or heuristic head designs, ignoring the high

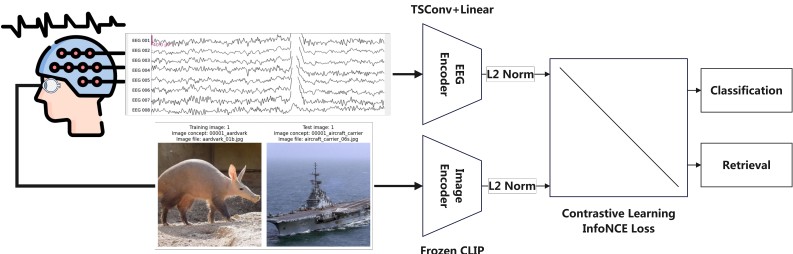

Figure 1: ARP Structure

noise and data scarcity of EEG, as well as the anisotropy of frozen vision spaces such as CLIP. We argue that understanding how much learnable capacity each side requires and how to regularize it is central to improving zero-shot generalization under strict 200-way evaluation.

## 3 METHODS

### 3.1 OVERVIEW

We address EEG–vision alignment with APR: a simple but effective recipe that constrains capacity on the EEG side while allowing only minimal, learnable calibration on the image side. Let $x \in \mathbb{R}^{C \times T}$ denote an EEG epoch (17 occipital–parietal channels, 100 time points) and $I$ a stimulus image. A temporal–spatial EEG encoder $f_\theta$ (TSConv) maps $x$ to a feature $\mathbf{h}_e = f_\theta(x)$. A frozen visual backbone $g(\cdot)$ (e.g., CLIP) produces an image embedding $\mathbf{v} = g(I)$. APR introduces two light-weight projectors into a shared space $\mathbb{R}^D$, but asymmetrically:

*EEG side (minimal head).* A single linear layer $P_E(\mathbf{h}_e) = W_E \mathbf{h}_e$ followed by $\ell_2$ normalization $\mathbf{z}_e = \text{norm}(P_E(\mathbf{h}_e))$. This "small EEG head" acts as a bottleneck/regularizer, discouraging the model from fitting trial-level noise.

*Image side (small learnable calibration).* A small projector $P_I$ applied to the frozen visual embedding, e.g., linear or a two-layer MLP with GELU (optionally low-rank LoRA (Hu et al., 2021)). We obtain $\mathbf{z}_i = \text{norm}(P_I(\mathbf{v}))$. The role of $P_I$ is to perform geometric alignment and scale calibration of the fixed image space with respect to the EEG space; APR deliberately avoids over-parameterizing $P_I$.

Training uses a symmetric InfoNCE objective over in-batch positives $(x, I)$ and negatives from the remaining batch items. With temperature $\tau$, the loss is

$$\mathcal{L} = \tfrac{1}{2}\Big[\mathcal{L}_{e \to i}(\mathbf{z}_e, \mathbf{z}_i) + \mathcal{L}_{i \to e}(\mathbf{z}_i, \mathbf{z}_e)\Big], \quad \mathcal{L}_{e \to i} = -\frac{1}{B} \sum_b \log \frac{\exp(\langle \mathbf{z}_e^{(b)}, \mathbf{z}_i^{(b)} \rangle / \tau)}{\sum_k \exp(\langle \mathbf{z}_e^{(b)}, \mathbf{z}_i^{(k)} \rangle / \tau)},$$

where $\langle \cdot, \cdot \rangle$ is the dot product after $\ell_2$-normalization (cosine similarity). We treat each repetition as an independent training instance; at test time, we average embeddings across 80 repetitions per concept to improve SNR, then perform 200-way zero-shot retrieval (EEG image gallery) and report closed-set Acc@K together with MRR and mRank↓.

Conceptually, APR enforces a nonsymmetric capacity allocation: (i) the EEG branch is strongly regularized (shallow projector) to enhance cross-concept generalization under high noise and low sample counts; (ii) the image branch retains just enough learnable degrees of freedom to reconcile the anisotropy and scale of the frozen visual space with the EEG space. Empirically (see ablations), enlarging the EEG head yields little benefit and can harm generalization, while over-flexible image heads reduce retrieval quality despite achieving lower training loss. Thus, APR offers a compact and reproducible training recipe, small head of EEG + small head of learnable image, that consistently improves the alignment of the EEG to the vision under constrained computation (see Fig. 1).

## 4 EXPERIMENTS AND RESULTS

### 4.1 DATASET AND PREPROCESSING

We conducted all experiments in the Things-EEG dataset (Gifford et al., 2022) and rely exclusively on the official preprocessed versions of the authors to ensure comparability and avoid redundant cleaning. For each of the 10 participants, a zipped archive provides two NumPy dictionaries: preprocessed_eeg_training.npy and preprocessed_eeg_test.npy, each containing three keys: preprocessed_eeg_data, ch_names, and times. The ch_names enumerate the 17 retained occipital–parietal electrodes; times records 100 sample points in seconds aligned to stimulus onset; and preprocessed_eeg_data store the epoched signals. The training partition is organized as a 4-D shape tensor [16,540 training image conditions × 4 repetitions of EEG × 17 channels × 100 time points], while the test partition has a shape [200 test image conditions × 80 repetitions × 17 channels × 100 time points]. In our pipeline, data are loaded as float32 and fed to the encoder as [repetition, channel(17), time(100)] (or [batch, C, T]). Following common practice, we treat each repetition as an independent training instance to exploit trial-level variability, whereas at evaluation, we compute the mean embedding across the 80 repetitions per concept to improve SNR and then perform 200-way zero-shot retrieval and closed-set classification. Because the data released already include artifact handling, filtering, and referencing, we do not apply additional preprocessing beyond repetition averaging at test time.

### 4.2 EXPERIMENTS AND DETAILS

All experiments run on Google Colab with a T4 GPU (15 GB VRAM), $\approx 12.7$ GB system RAM, and $\approx 235$ GB disk available. The code is implemented in PyTorch with CUDA acceleration. We impose identical data splits and hyperparameter schedules across ablations to ensure fair comparison.

The EEG branch uses the NICE temporal–spatial conv encoder with a single linear projection to the shared dimension $D$ followed by normalization $\ell_2$. APR instantiates asymmetric capacity by pairing this minimal EEG head with a small, easily learnable image head. Unless noted otherwise, the image projector is linear (I:linear); ablations consider MLP-2 (linear–GELU–linear, hidden=768, dropout=0.1), MLP-2-Res (with residual to the output) and LoRA adapters (rank=8, base linear frozen). Symmetric/EEG-heavier variants (E:MLP-2 / E:MLP-2-Res / E:LoRA with I:linear) are also reported to diagnose over-parameterization on the EEG side.

We report Acc@1/5/10 on the 200-way classification protocol; for retrieval we report Top-K, MRR, and mRank ↓. As observed in previous work and in our experiments, Acc@K in the closed-set protocol matches Top-K retrieval with the same gallery.

### 4.3 PERFORMANCE COMPARISON

We report Top@1/Top@5, MRR, mRank↓, Acc@1 and Acc@5 for each of the 10 participants under three temporal regimes: an early visual window ($\approx$0.08–0.46 s), a later window ($\approx 0.14$–0.80 s) and the full epoch. A consistent pattern emerges across metrics: performance is highest in the early and full settings and degrades markedly in the later window, with mRank showing the complementary improvement (lower is better). This indicates that discriminative information for EEG-to-vision alignment is concentrated in the early post-stimulus period that overlaps well with feedforward visual processing, whereas extending the window into later activity adds noise relative to signal for zero-shot recognition. Intersubject variability is present, yet bounded, e.g., subjects 09-10 tend to score near the top on all metrics, while the ordering between subjects is largely preserved between Top@K, MRR, mRank, and Acc@K closed set. In general, the figure corroborates the temporal location of useful EEG signals and the robustness of APR between participants and evaluation metrics (see Fig. 2).

Topographic visualizations of Grad-CAM (Selvaraju et al., 2020) weights computed on the EEG encoder with respect to the image–EEG similarity score for ten held-out test concepts (titles show the concept string; bottom labels indicate the concept id and category). For each concept, Grad-CAM vectors were obtained in the encoder projection layer, averaged over a fixed number of repetitions, normalized to $\ell_2$ and interpolated to a 10-10 scalp layout on the 17 occipital-parietal channels (hotter colors = higher attribution; contours indicate isoweight lines). The maps consistently highlight

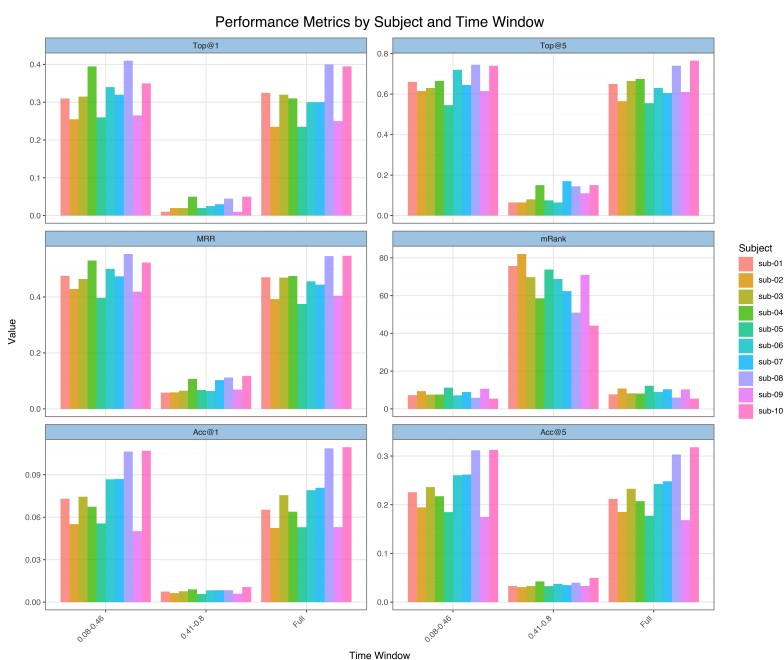

Figure 2: Subject-wise performance across temporal windows

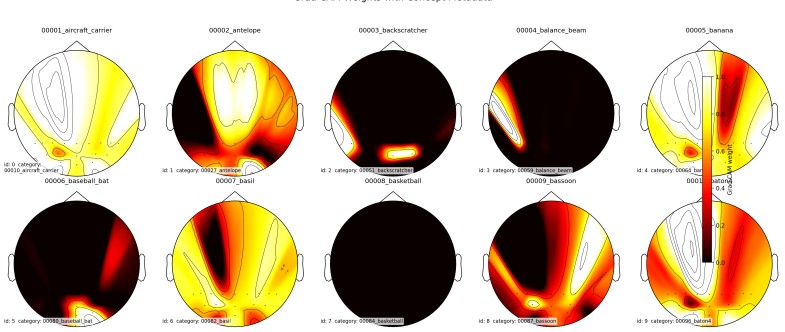

Figure 3: Subject-01's Grad-CAM Weights with Concept Metadata

posterior regions associated with visual processing (occipital/occipito-temporal), while exhibiting concept-specific spatial patterns (e.g., lateralized foci or ventral–dorsal shifts) that are stable across repetitions. These qualitative results suggest that the learned alignment under APR concentrates the attribution in physiologically plausible visual areas and varies meaningfully with semantic content, complementing our quantitative retrieval/classification gains. We emphasize that the plots are descriptive rather than inferential and are meant to illustrate encoder attention rather than claim strict neuroanatomical localization (see Fig. 3).

Table 1: Overall accuracy (%) of 200-way zero-shot classification: top-1 and top-5

| Method | Subject 1 | | Subject 2 | | Subject 3 | | Subject 4 | | Subject 5 | | Subject 6 | | Subject 7 | | Subject 8 | | Subject 9 | | Subject 10 | | Ave | |
|---|---|---|---|---|---|---|---|---|---|---|---|---|---|---|---|---|---|---|---|---|---|---|
| | top-1 | top-5 | top-1 | top-5 | top-1 | top-5 | top-1 | top-5 | top-1 | top-5 | top-1 | top-5 | top-1 | top-5 | top-1 | top-5 | top-1 | top-5 | top-1 | top-5 | top-1 | top-5 |
| BraVL | 6.1 | 17.9 | 4.9 | 14.9 | 5.6 | 17.4 | 5.0 | 15.1 | 4.0 | 13.4 | 6.0 | 18.2 | 6.5 | 20.4 | 8.8 | 23.7 | 4.3 | 14.0 | 7.0 | 19.7 | 5.8 | 17.5 |
| NICE | 12.3 | 36.6 | 10.4 | 33.9 | 13.1 | 39.0 | 16.4 | 47.0 | 8.0 | 26.9 | 14.1 | 40.6 | 15.2 | 42.1 | 20.0 | 49.9 | 13.3 | 37.1 | 14.9 | 41.9 | 13.8 | 39.5 |
| NICE-SA | 13.3 | 40.2 | 12.1 | 36.1 | 15.3 | 39.6 | 15.9 | 49.0 | 9.8 | 34.4 | 14.2 | 42.4 | 17.9 | 43.6 | 15.2 | 50.2 | 14.4 | 38.7 | 16.0 | 42.8 | 14.7 | 41.7 |
| NICE-GA | 15.2 | 40.1 | 13.9 | 40.1 | 14.7 | 42.7 | 17.6 | 48.9 | 9.0 | 29.7 | 16.4 | 44.2 | 14.9 | 43.1 | 20.3 | 52.1 | 14.1 | 39.7 | 19.6 | 46.7 | 15.6 | 42.8 |
| **ARP (ours)** | 34.0 | 64.0 | 22.5 | 57.5 | 32.5 | 65.5 | 31.0 | 67.5 | 23.5 | 54.5 | 30.0 | 62.5 | 30.5 | 60.0 | 39.0 | 73.5 | 22.5 | 60.5 | 39.0 | 76.5 | **30.5** | **64.2** |

*Noted:* Subject dependent - train and test on one subject.

## 4.4 ABLATIONS

Across three metrics, Top-1 accuracy (200-way closed set, equivalent to zero shot retrieval), retrieval MRR, and mean rank (mRank ↓), the same pattern emerges and supports APR. The best regime is a small EEG head plus a small learnable image head (E:linear / I:linear). With the image head fixed to linear, increasing the EEG-side capacity to MLP-2, MLP-2-Res, or LoRA brings no gain and slightly worse performance, consistent with overfitting to noisy, low-sample EEG. In contrast, with the EEG head fixed to linear, making the image head more complex (I:MLP-2 / I:MLP-2-Res / I:LoRA) consistently degrades Acc@1, reduces MRR, and worsens mRank, indicating that frozen visual features require only minimal learnable calibration for geometric alignment and scale normalization.

Importantly, training dynamics reveal a decoupling between optimization loss and generalization: configurations that add image side capacity or introduce EEG side LoRA reach lower InfoNCE loss ($\approx 3.0$) yet underperform on test metrics, while the APR regime (E:linear / I:linear) converges to a higher training loss ($\approx 4.3\text{-}4.4$) but yields the best retrieval and classification, suggesting stronger regularization against trial-level EEG noise. In particular, E:linear / I:LoRA exhibits both higher loss and poor test metrics, pointing to an optimization–alignment mismatch when the image projector is overly flexible. Collectively, these results confirm the core claim of APR: asymmetric capacity - a minimal EEG projector paired with a small learnable image projector - provides the most reliable alignment and retrieval order of EEG to vision, while lower training loss is not predictive of better zero-shot performance in this noisy low sample setting (see Fig. 4).

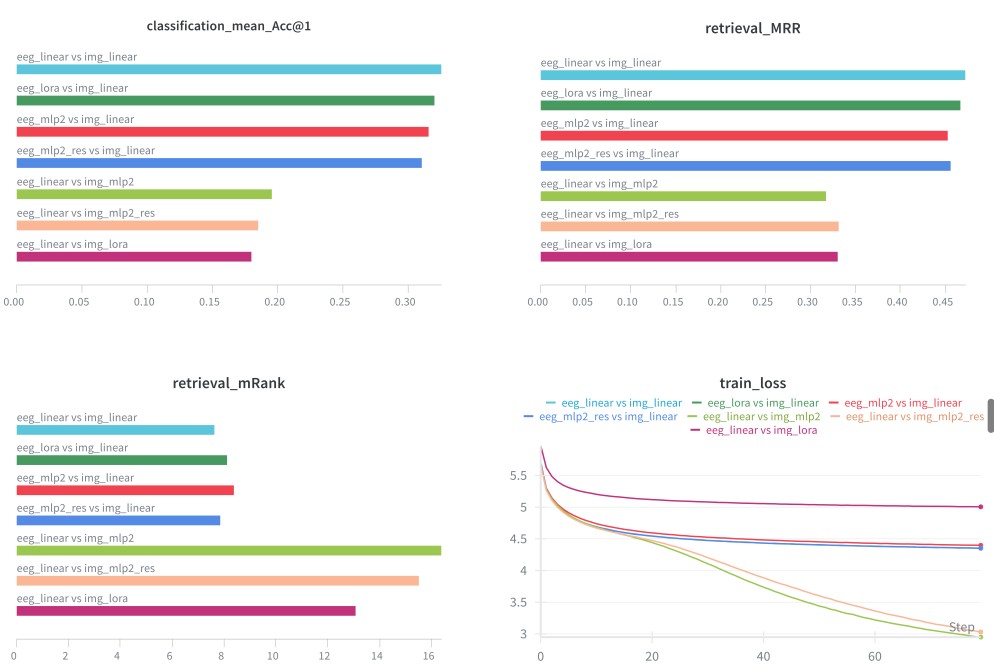

Figure 4: Ablation on ARP with Top-1, Retrieval MRR, mRank and train loss.

## 5 DISCUSSION AND CONCLUSION

In this work, we revisit a largely overlooked design choice in contrastive EEG-vision learning: projection head symmetry. Contrary to conventional practice, our results show that projection heads on the two sides should not be configured identically. Under a fixed contrastive framework with frozen CLIP embeddings, simplifying the EEG head to a single linear layer with L2 normalization yields large improvements in zero-shot generalization (200-way Top-1: 13.8% → 30.45%). In con-

trast, removing learnable capacity on the vision side (e.g., identity or fixed mapping) leads to severe degradation in retrieval and classification.

We attribute this asymmetry to two factors:

1. Noise amplification on the EEG side. EEG data are scarce and noisy; deeper EEG heads tend to memorize trial-level noise and image IDs. A shallow linear head with normalization acts as a bottleneck regularizer, restricting variability to the encoder backbone.

2. Alignment on the vision side. Frozen CLIP embeddings remain geometrically misaligned with EEG features. A small but learnable projection head on the vision side absorbs the necessary calibration; removing it forces the EEG side to overcompensate, often collapsing training.

Comprehensive experiments confirm these findings.

- Vision-side heads must retain minimal learnability; fixed mappings collapse retrieval accuracy.
- APR is sensitive but tunable with respect to contrastive temperature and batch size ($\tau \approx 0.05$, batch $\geq 512$).
- Theveraging of embeddings between trials improves the SNR and stability.
- The gains hold consistently across time windows, subjects, and random seeds.

*Takeaway.* Projection asymmetry matters. Alignment and calibration should be handled primarily by the vision side, while denoization and generalization should be regularized on the EEG side.

*Conclusion.* We introduce APR, a simple modification that substantially improves zero-shot recognition from EEG to vision without altering the backbone or training paradigms. The APR gains are based on clear geometric and statistical principles: stable modalities should carry alignment, noisy modalities should carry regularization. We hope that APR can serve as a reusable recipe for future EEG decoding research, including diffusion-based reconstruction and multimodal integration, and inspire systematic study of structural regularization in cross-modal alignment.

## 6 ETHICS STATEMENT

This work uses only the publicly released Things-EEG dataset (Gifford et al., 2022). We did not collect new human data. According to the dataset documentation, recordings were obtained under institutional review and informed consent; data are de-identified. We followed the license terms and used the preprocessed partitions without any attempt to re-identify participants or to infer sensitive attributes. All analyses are aggregate and report per-subject metrics only for scientific reproducibility.

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
